# Modeling of EDM Process Flushing Mechanism

**DOI:** 10.3390/ma16114158

**Published:** 2023-06-02

**Authors:** Timur Rizovich Ablyaz, Evgeny Sergeevich Shlykov, Karim Ravilevich Muratov

**Affiliations:** Department of Mechanical Engineering, Perm National Research Polytechnic University, Perm 614000, Russia; kruspert@mail.ru (E.S.S.); karimur_80@mail.ru (K.R.M.)

**Keywords:** functionalmaterial titanium VT6, electrical discharge machining, flushing sludge

## Abstract

The study was performed to determine the optimum flushing condition for electrical discharge machining (EDM) of functional material titanium VT6 obtained by plasma cladding with a thermal cycle. Copper is used as an electrode tool (ET) to machine functional materials. The optimum flushing flows are analyzed theoretically by using ANSYS CFX 20.1 software which is also validated by an experimental study. It was observed that while machining the functional materials to adepth of 10 mm or more, the turbulence fluid flow dominates when nozzle angles are 45° and 75°, consequently drastically affecting the quality of flushing and the performance of the EDM. For the highest machining performance, the nozzles should be at an angle of 15° relative to the tool axis. The optimum flushing at deep hole EDM process minimizes the occurrence of debris deposition on tool electrodes, thus facilitating stable machining of functional materials. The adequacy of the obtained models was confirmed experimentally. It has been established that EDM of a hole with a depth of 15 mm, an intense accumulation of sludge, is observed in the processing zone. There arebuild-ups exceeding 3 mm in cross-section after EDM. This build-up leads to a short circuit and a reduction in surface quality and productivity. It has been proven that not correct flushing leads to intensive wear of the tool and a change in its geometric shape and, accordingly, to a decrease in the quality of EDM.

## 1. Introduction

The development of the machine-building structure is associated with the emergence of new, fixed development of technologies and equipment. Obtaining considered layers of materials with controlled macro- and microstructure makes it possible to form the required physical and mechanical properties of products with operating conditions. As a result, such products are ubiquitous and have a mechanical composition, heterogeneous chemical composition and increased hardness.

It is well known that the conventional machining of functional materials is complicated. Thus, electrophysical processing, such as EDM, is an efficient method used to process functionalmaterials. However, EDM of functional materials is viewed to be a challenging process. This is dueto the fact that the cooling in the EDM processing zone is inadequate [1,2,3].

EDM of functional materials is presented by various researchers [4,5,6], and it is revealed that a product made of functional materials comes under the influence of electrical impulses during EDM. The plasma channel formed between electrodes (tool electrode and functional materials) has a temperature of about 9000–9500 °C [1,2,3]. Due to this high-temperature zone in the EDM process, the phase transition of functional material and deposition of molten pieces ofET that solidify upon cooling deteriorates the properties of functional materials. As a result, these molten residues have a negative impact on the quality of the workpiece and the performance of the EDM [7,8].

Accumulation deposition of erosive residues in the machining zone is due to poor flushing within the space between the ET and the workpiece, specifically when deep holes are to be obtained or slots and key grooves. This phenomenon leads to the appearance of secondary dendritic structures on the surface of the ET and the workpiece, thus decreasing the quality and productivity of the functional materials products [7].

It has been established that during the EDM of functional materials products, the movement of debris residues can be achieved by forming and moving gas bubbles in the processing zone [8,9,10,11]. The dielectric is a viscous fluid, and the EDM slurry moves in the shell of the gas bubble. The works [8,9,10,11,12,13] demonstrate the movement of erosive sludge debris in interelectrode space. However, in these works, there are no practical recommendations for increasing the productivity and efficiency of functional materials products.

The heating of the processed material causes the thermal decomposition of the workpiece and the dielectric medium [14,15]. This dielectric medium is in a state of motion and constant circulation. Thus, it leads to the cooling of the ET and the workpiece material. The vapor formed during EDM flows becomes turbulent and breaks up into small fractions. Each part can condense into a liquid, and as a result, into a solidstate.

The molten metal promotes spheroidization and dendritic segregation. The vapors of the dielectric fluid reduce the cooling rate of the drop. Furthermore, at low EDM energy, the dielectric fluid and content of metal vapors decrease. This decreases the number of particles and has a smaller average size. The flow of the material and the vapors of the working fluid increases with the increase in the values of pulsed energy [14,15]. The movement of sludge/debris particles is turbulent, and there is a collision between them. Thus, cracks and dents form on the surface of the particles of this sludge, and an inclusion structure also appears. EDM slurry formation affects the stability of the EDM process and productivity. Increasing the productivity of the EDM process is achieved by increasing the pulse energy and by intensifying the removal of erosion products from the interelectrode gap. Productivity increases with effective flushing and intensive removal of eroded particles of functional materials and ET from the gap. Flushing brings clean dielectric into the gap and cools the ET and functional materials. The deeper the hole, the more difficult to ensure proper flushing of the machined area. Thus, increasing processing time and reducing performance. Under certain processing conditions, the eroded particles are welded/deposited onto the product of functional material. This leads to uneven processing and reduced performance or even to its cessation.

The significance of flushing is vital in EDM of deep holes. Insufficient flushing diminishes material removal efficiency. The material that remains in the hole or machining zone is remelted in the next pulse and welded onto the electrode surface resulting in poor machining [15,16].

The intensification of flushing during EDM in deep and narrow cavities contributes to an increase in the material removal rate. It was found that flushing and ET jump maintain material evacuation rate after a discharge (i.e., pulse-off time) [17,18,19,20]. The electrode jump speed affected the distribution of eroded particles, and the jump amplitude affected the amount of fresh dielectric.

It is shown that the pressure drops of dielectric fluid enhance with an increase in hole depth [18,19,20,21,22,23,24]. This pressure drop is about 15% at 25 mm hole depth, and the accumulation of a higher concentration of eroded material was also observed at the corner of the machined hole [23,24,25,26]. In work [23,24,25], the experiments were carried out using electrical discharge machining with side flushing and multi-aperture flushing to improve the machining performance and surface integrity. The machine parameters were pulse ontime, pulse off-time, current, and electrode rotation. However, these works do not take into account the influence of the angle of inclination of the nozzles relative to the depth on the processing performance and the magnitude of the liquid pressure at the processing site. Figure 1 shows the schematic of side flushing arrangements in the EDM process.

The researcher studies the side flushing mechanism in EDM; however, the efficiency of flushing in EDM deep holes in functional materials with intricate geometry has not been practically studied in detail. Existing models in the EDM of functional materials can be analyzed using finite element analysis software systems. The study aims to develop a theoretical model of a deep hole flushing system for EDM of functional materials and validation of the model with an experimental study.

## 2. Materials and Methods

In this experimental study, a copper M1(Med Prom Star, Moscow, Russia) cylinder diameter10 mm was chosen as an ET. The workpiece was made offunctional material titanium VT6 obtained by plasma cladding with a thermal cycle. This blank provides high reproducibility and defect-free layer income. In the usual form of restoration, the restoration of the fatigue strength of parts is 10–15%. Workpiece hardness 45–55 HRC. The blanks are obtained in this way for the restoration of working structures, and production in aggressive environments, as well as in the oil industry and mechanical engineering. The workpiece was processed in EDM oil of grade IPOL SEO 450 (IPOL Lubricants, Mumbai, India) on an Electronica Smart CNC copy-piercing EDM machine (Electronica Machine Tools, Pune, India). The surface roughness after EDM was evaluated using a Lext OLS4000 laser scanning microscope (Olympus Corporation, Tokyo, Japan) on a 3D image model obtained using the 3D Roughness Reconstruction software module (Olympus Corporation, Tokyo, Japan). To build 3D surface models, we used optical sections obtained during X-Y-Z scanning of the surface areas of interest. Scanning was performed at a magnification of ×100; the scanning step along the Z axis was from 2 μm. A semiconductor laser with a wavelength of 405 nm was used as a light source for scanning. An Olympus GX 51 light microscope (Olympus Corporation, Tokyo, Japan) at 100× and 50× magnification was used to visually evaluate texturing results and to measure dimensions ET.This oil provides stable cooling of the EDM process during power surges, as well as electrical insulation, which helps to avoid short circuits. The processing parameter selected is constant voltage U = 50 V, pulse on time (Ton) 100 μs, and current (I) 10 A [4,5,20]. ANSYS CFX 20.1 (Canonsburg, PA 15317, USA) software was used for the theoretical flow simulation of the dielectric (Transformer oil). The ANSYS CFX package is a universal CFD (Computational Fluid Dynamics—Computational Fluid Dynamics) system. The main areas of application of this package are hydro- and gas-dynamic processes, and chemical kinetics. By means of this package, such problems are solved as visualization of the flow of liquids and gases around solid bodies, combustion and heat transfer processes, heat transfer processes, ventilation and distribution of air and gases, and radiative heat transfer. The dielectric temperature is set as standard (i.e., equal to 25 °C). For all cases, the pressure was 2.1 kg/cm^2^ (0.205 MPa). The modeling of the flow’s distribution was carried out at three values of the processing depth of the hole (i.e., 2 mm, 10 mm, and 15 mm), as well as at three values of the inclination angle of the nozzles with respect to tool axis (i.e.,15°, 45°, and 75°).

The purpose of the simulation is to obtain a theoretical model for the distribution of working fluid flows in the machining zone, subjected to a change in the angle of flushing. To attain an objective, it is necessary to specify the geometry of the concerned area, define the boundary conditions, analysis of the model for the processing depth of 2 mm, 10 mm, and 15 mm, and the location of the nozzles 15°, 45°, and 75° relative to the tool axis (Figure 2).

The experimental part carried out in works [4,5,6] shows that products made of functional materials during EDM are prone to sludge fusing on the treated surface. This was owing to the irrational location of the flushing nozzles and the formation of turbulence in the machining zone.

Modeling is performed after defining the boundary conditions, such as part walls, ET, and flush nozzles. Geometrical limits are similar for 2 mm, 10 mm, and 15 mm machining depth, while the angle of the nozzle changes (Figure 3).

The computational grid is shown in Figure 4. A single voxel’s minimum and maximum values are set to build the grid: min—1 mm, max—5 mm. The authors perform the same condition for other cases. The nozzles will operate at the uniform pressure (0.205 MPa) and angle selected relative to the tool axis.

The mesh has been taken at minimum values in the processing area and boundary areas for high accuracy in simulation. The single-phase oil working fluid flow is modeled using the standard turbulence model (Figure 5).

The oil flow pattern was collected using an enlarged image of the electrode cross-section and simplified to reduce computation time. The number of elements of the tetrahedral grid varies from 7.8 million to 6.4 million elements in the workpiece hole. The calculations are carried out in the ANSYS FluidFlow module.

## 3. Results and Discussion

### 3.1. Theoretical Modeling

Based on the data obtained, it was found that the influence of the angle of the nozzles on the flushing efficiency was not significant when processing a functional material sample to a depth of 2 mm. Figure 6, Figure 7 and Figure 8 show that the laminar flow of the fluid dominates.

While processing to the depth of 10 mm, it was seen that the laminar movement of the working fluid prevails for the nozzle located at an angle of 15°. However, it is noted that for nozzles located at angles of 45° and 75°: turbulence is formed in the inter-electrode gap and entails a slight decrease in pressure. As a result, the sludge/debris removal from the machining zone is difficult, as shown in Figure 9, Figure 10 and Figure 11.

Figure 12, Figure 13 and Figure 14 show that at a working depth of 15 mm for a nozzle located at an angle of 15°, the laminar movement abruptly turns into turbulence. In the processing zone, the flows of the two nozzles collide. Turbulent motion dominates completely. It has been observed that when processing holes of a given depth (15 mm) and above, the nozzle’s location at an angle of 45° and 75° relative to the tool axis is inappropriate. This is caused by high flow turbulence and loss of transformer oil (dielectric) pressure in the processing zone.

From the results, we can conclude that turbulent movement prevails when the nozzles are located at 45° and 75°. This entails a reduction in dielectric pressure. The pressure value for the nozzle at 75° did not exceed 0.14 MPa (Figure 14), while the nozzle at 15° provided a rational pressure in the treatment zone from 0.1 MPa to 0.2 MPa (Figure 12).

It is shown that the location of the nozzle at 75° for processing holes deeper than 10 mm reduces the pressure in the processing zone. For processing holes deeper than 15 mm, the location of the nozzles at 75° critically affects the pressure and speed of the working fluid and the time of the removal of eroded particles from the machining zone. 

### 3.2. Experimental Studies

An experimental study was carried out to validate the theoretical modeling of EDM performance on functional materials. Figure 15 depicts the comparison of the performance of the EDM process at various nozzle angles and variable machining depths.

It was concluded that when processing a hole up to the depth of 2 mm, the value of the angle of inclination of the flushing nozzle does not affect the performance of the EDM.

The influence of nozzle inclination angle was manifested during the EDM to a depth of 10 mm and 15 mm. A reduction in the productivity of the EDM was observed due to difficulty in flushing the interelectrode space. Thus, the angle of nozzle inclination should be taken into account for processing holes with a depth of 10 mm or more. For adequate machining, eroded particles/debris must be removed from the gap. During the experimental analysis, it was observed that when the processing holes were to a depth of 15 mm, sludge/debris adhered to the ET surface. Therefore, the closure of the EDM process has also occurred.

The study of the electrode surfaces at an angle of inclination of 45 degrees was carried out to confirm the theoretical models. Depths of processing are chosen as 2 mm and 15 mm. The scheme of installation of electrodes and flushing nozzles is shown in Figure 16.

Experimental equipment and processing modes are described in Section 2. Stable sparking was observed in the process of processing the workpiece to a depth of 2 mm. The process proceeded without the occurrence of voltage and current surges. ET after treatment is shown in Figure 17.

Analysis of Figure 17 shows uniform ET wear. The working area of the electrode changed its roughness during processing. The formed roughness is stable over the entire working part of the electrode. There are no traces of sticking of the material of the workpiece. The slope is flat. The growths are absent. The side zone of the electrode is characterized by a slight change in the surface structure. The surface is uniform. The mathematical model is confirmed. Flushing ensures that the sludge is washed out of the treatment area.

The ET analysis (Figure 18) shows uneven wear on its running surface. Traces of build-up are visible at the end of the ET. The ET surface is not a stable profile. This phenomenon is explained by the results of mathematical modeling. The sludge is concentrated in the center of the interelectrode gap. As a result, there are additional secondary discharges. This results in uneven ET. The lateral part of the ET is characterized by an intensely modified structure over the entire working height. The effect of poor flushing of the interelectrode gap has an effect. The concentration of erosion products contributed to the occurrence of side discharges. A three-dimensional image of the surface of the machined hole to a depth of 2 mm is shown in Figure 19.

Analysis of Figure 19 showed that the surface of the machined workpiece is characterized by a uniform relief. There are no protrusions orirregularities. It is noted that during the processing, there is no sticking of the material. The shape of the electrode during processing was stable. The data obtained fully confirm the results of the theoretical model on the stability of washing the treatment zone under these modes. Figure 20 shows a three-dimensional image of the machined surface after piercing to a depth of 15 mm. For the convenience of three-dimensional scanning, a section 5 mm deep was cut out of the processed workpiece.

Analysis of Figure 19 showed that the surface of the machined workpiece is uneven. There are irregularities in the form of growths formed as a result of the melting of the tool electrode material and the concentration of sludge in the processing zone. The height of irregularities in some sections exceeds 3 mm. During the processing, the accumulated sludge caused a short circuit. As a result, sludge build-up on the part surface and ET were present. The concentration of the sludge is fully confirmed by the conditions of poor flushing. These phenomena are confirmed by the obtained mathematical models. An analysis of the experimental data showed the adequacy of the developed approach to modeling the process of flushing the interelectrode gap during EDM.

Furthermore, secondary discharges in the machining zone were also observed, thus causing the machining process to be discontinued. The obtained experimental data confirms the results of theoretical modeling.

## 4. Conclusions

The study was conducted to determine the rational flushing condition for EDM of functional materials. As a result, the following conclusion has been drawn: A theoretical model has been developed and successfully demonstrates the optimum location of the nozzle for machining functional materials at different depths;It has been established that at a machining depth of 2 mm, the location of the nozzles does not affect the quality of flushing and the performance of EDM functional materials. The laminar fluid flow dominates in this depth of machining;It has been shown that when the EDM of functional materials to a depth of 10 mm and 15 mm, the location of the nozzles drastically affects the quality of flushing and the performance of the EDM. The highest performance (MRR) value is achieved when the nozzles are located at an angle of 15°. Moreover, turbulent fluid flow occurs when nozzle angles are 45° and 75°;The sticking of sludge on the surface of the ET and the occurrence of a short circuit have been experimentally confirmed. This leads to the instability of the EDM offunctional material (titanium VT6 obtained by plasma cladding with a thermal cycle). For holes with a depth of 15 mm, the location of the nozzles at 75° critically affects the pressure, speed of the working fluid, and the removal of eroded particles from the machining zone, thus reducing machining performance;The conducted experimental studies have shown the adequacy of the obtained mathematical models. It has been established that an intense accumulation of sludge in the processing zone appears when processing a hole with a depth of 15 mm. The growth occurs. It exceeds 3 mm in cross-section. This build-up leads to the occurrence of a short circuit and the appearance of secondary discharges and a decrease in the quality of the machined surface, and a decrease in productivity. Uneven flushing leads to intensive wear of the tool and changes in its geometric shape, which reduces the quality of processing.

## Figures and Tables

**Figure 1 materials-16-04158-f001:**
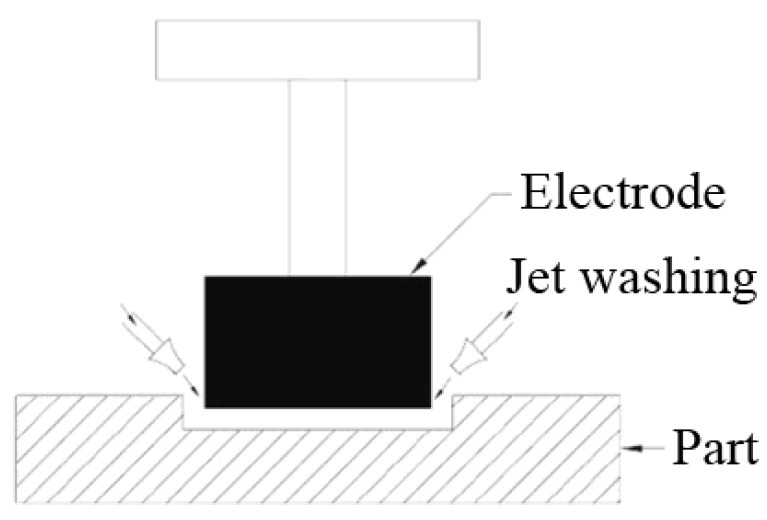
Scheme of jet or side washing.

**Figure 2 materials-16-04158-f002:**
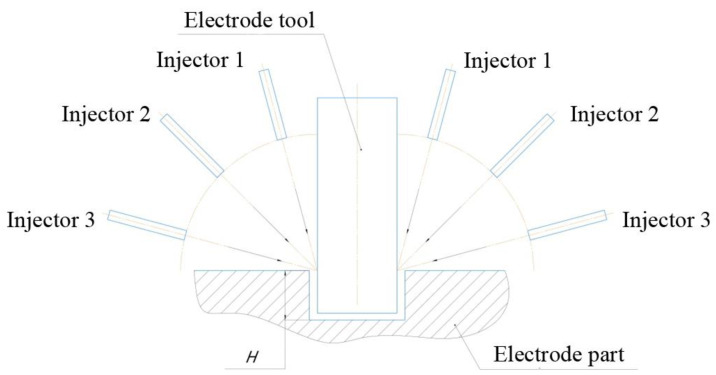
Schematic diagram processing model, where H is the depth of processing.

**Figure 3 materials-16-04158-f003:**
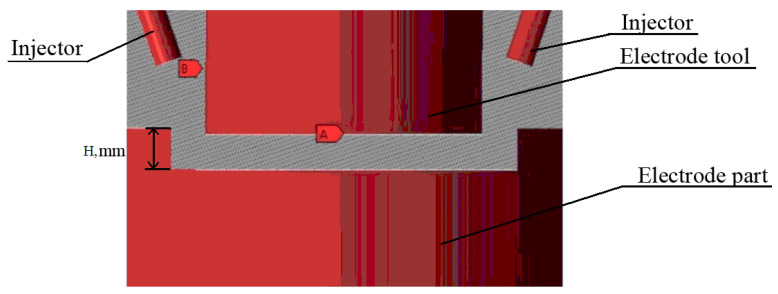
Processing model, where H is the depth of processing.

**Figure 4 materials-16-04158-f004:**
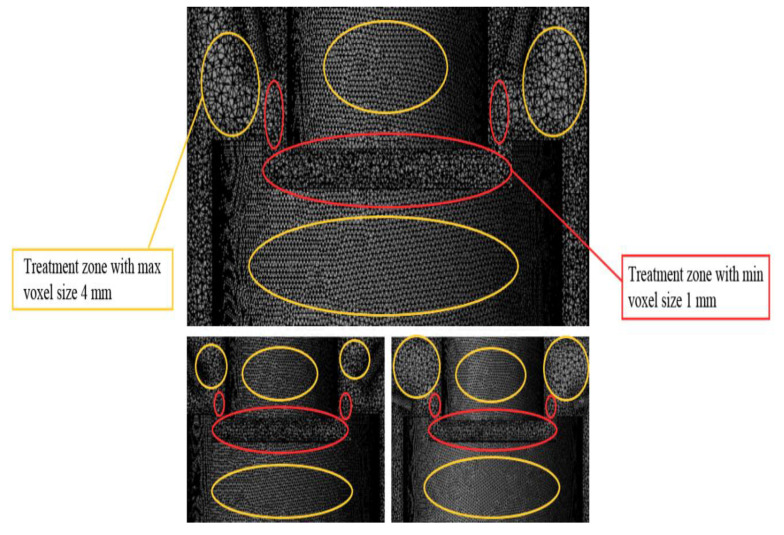
The computational grid.

**Figure 5 materials-16-04158-f005:**
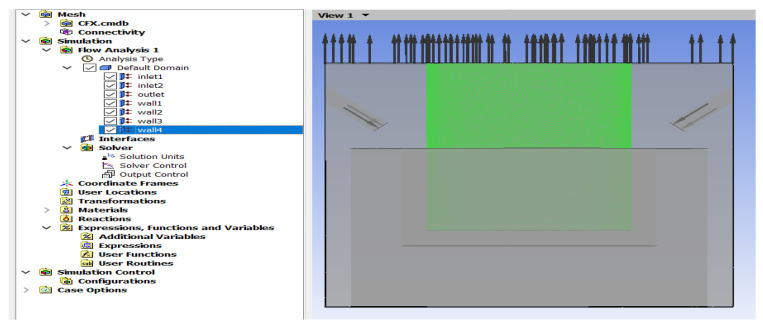
Calculation construction tree with the final model of the CFXPRE module.

**Figure 6 materials-16-04158-f006:**
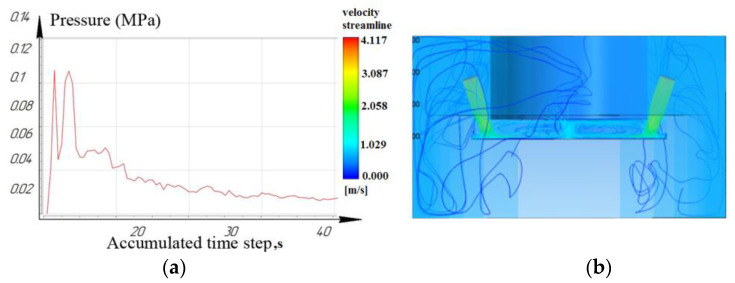
Depth 2 mm, nozzle angle 15°, (**a**) calculation of the pressure of the working fluid, (**b**) flow distribution models.

**Figure 7 materials-16-04158-f007:**
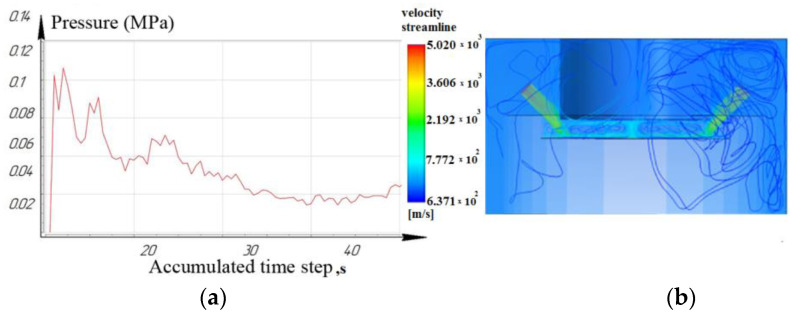
Depth 2 mm, nozzle angle 45°, (**a**) calculation of the pressure of the working fluid, (**b**) flow distribution models.

**Figure 8 materials-16-04158-f008:**
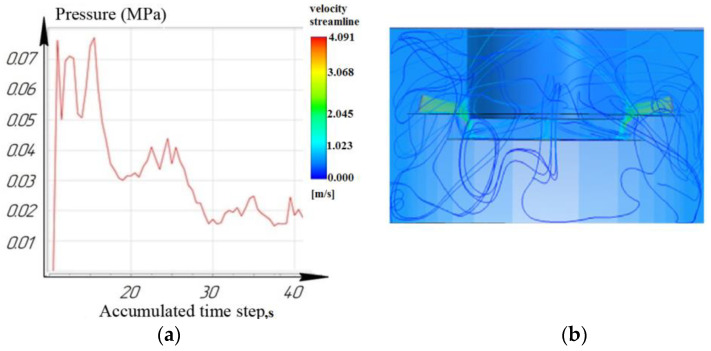
Depth 2 mm, nozzle angle 75°, (**a**) calculation of the pressure of the working fluid, (**b**) flow distribution models.

**Figure 9 materials-16-04158-f009:**
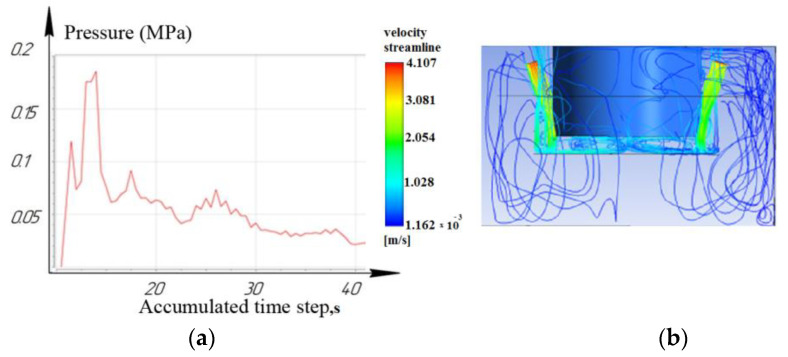
Depth 10 mm, nozzle angle 15°, (**a**) calculation of the pressure of the working fluid, (**b**) flow distribution models.

**Figure 10 materials-16-04158-f010:**
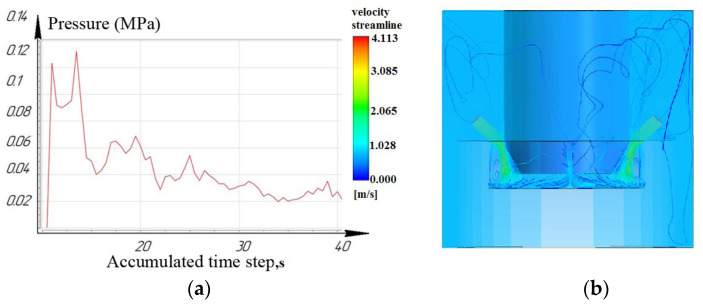
Depth 10 mm, nozzle angle 45°, (**a**) calculation of the pressure of the working fluid, (**b**) flow distribution models.

**Figure 11 materials-16-04158-f011:**
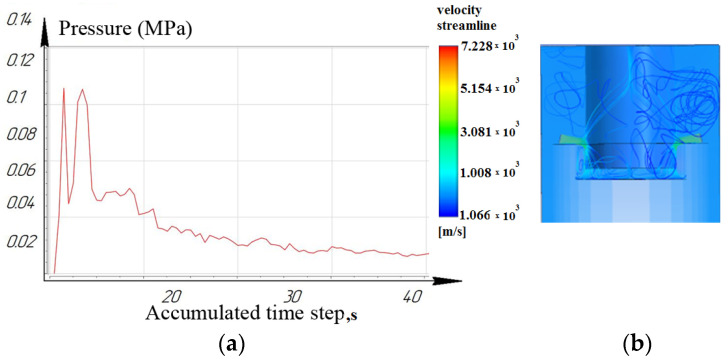
Depth 10 mm, nozzle angle 75°, (**a**) calculation of the pressure of the working fluid, (**b**) flow distribution models.

**Figure 12 materials-16-04158-f012:**
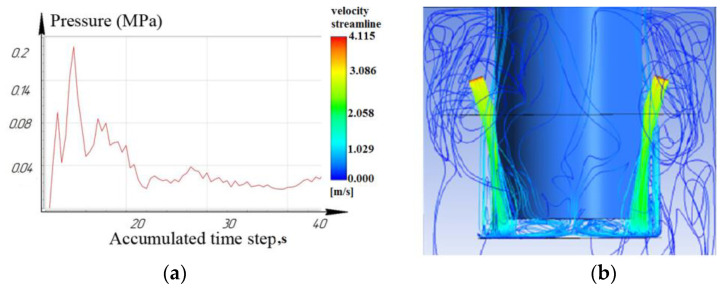
Depth 15 mm, nozzle angle 15°, (**a**) calculation of the pressure of the working fluid, (**b**) flow distribution models.

**Figure 13 materials-16-04158-f013:**
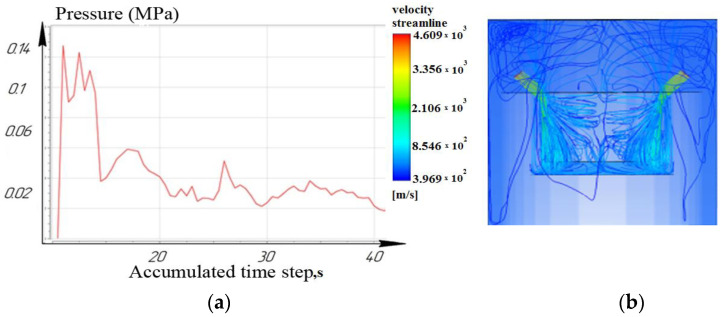
Depth 15 mm, nozzle angle 45°, (**a**) calculation of the pressure of the working fluid, (**b**) flow distribution models.

**Figure 14 materials-16-04158-f014:**
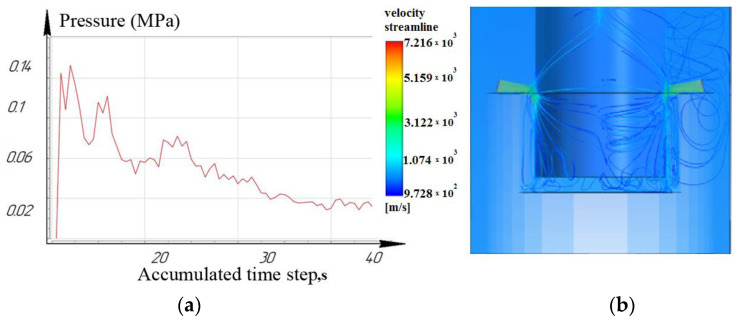
Depth 15 mm, nozzle angle 75°, (**a**) calculation of the pressure of the working fluid, (**b**) flow distribution models.

**Figure 15 materials-16-04158-f015:**
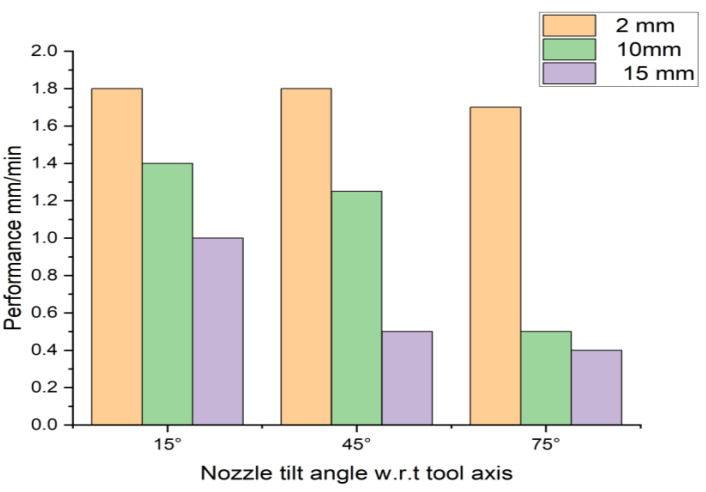
Performance values (Q = Material Removal Rate).

**Figure 16 materials-16-04158-f016:**
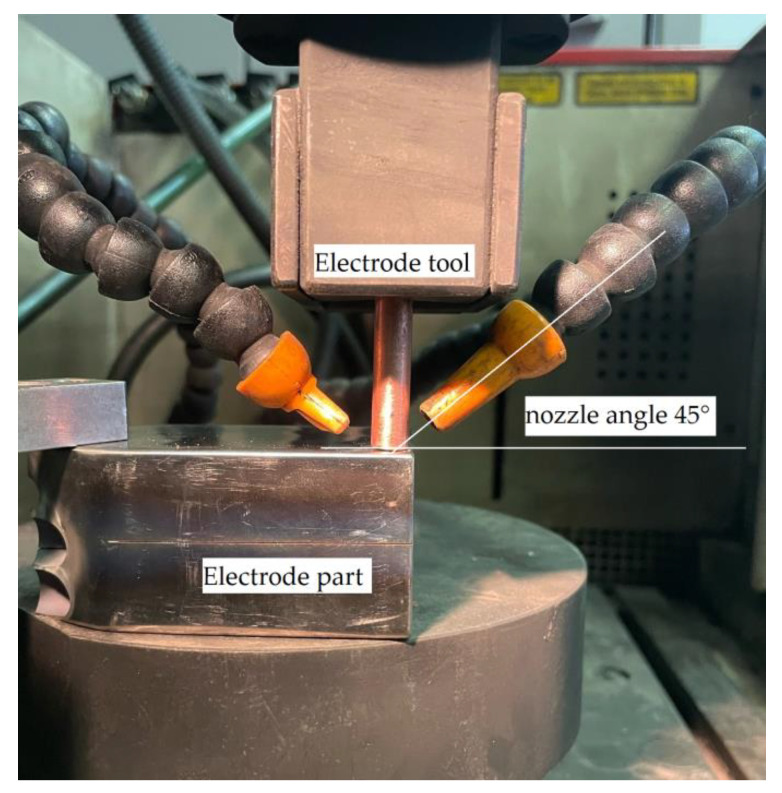
Scheme of the experimental setup.

**Figure 17 materials-16-04158-f017:**
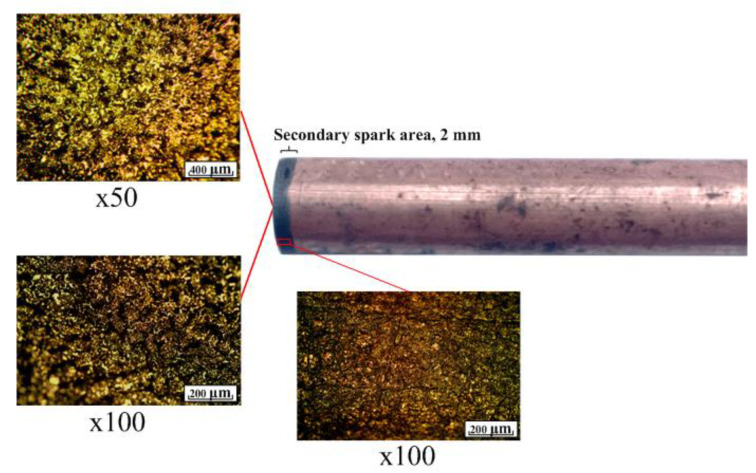
ET after machining to a depth of 2 mm.

**Figure 18 materials-16-04158-f018:**
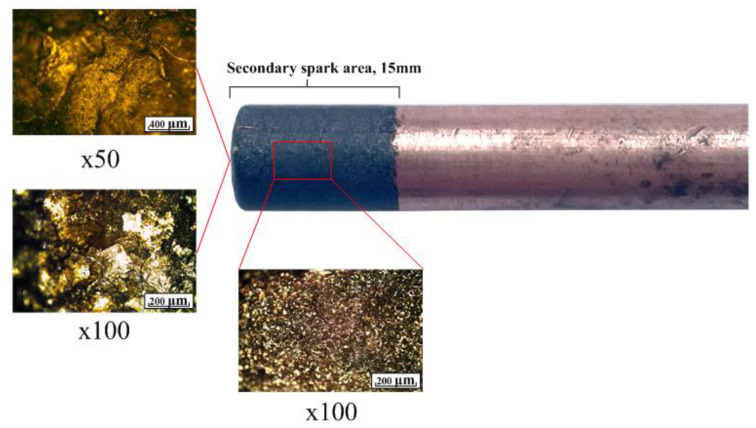
ET after machining to a depth of 15 mm.

**Figure 19 materials-16-04158-f019:**
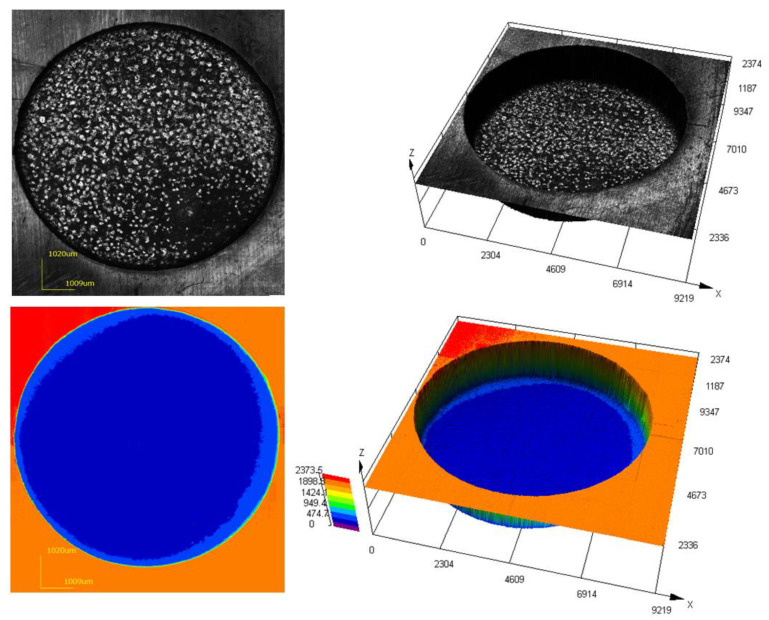
Three-dimensional image of the surface after processing to a depth of 2 mm.

**Figure 20 materials-16-04158-f020:**
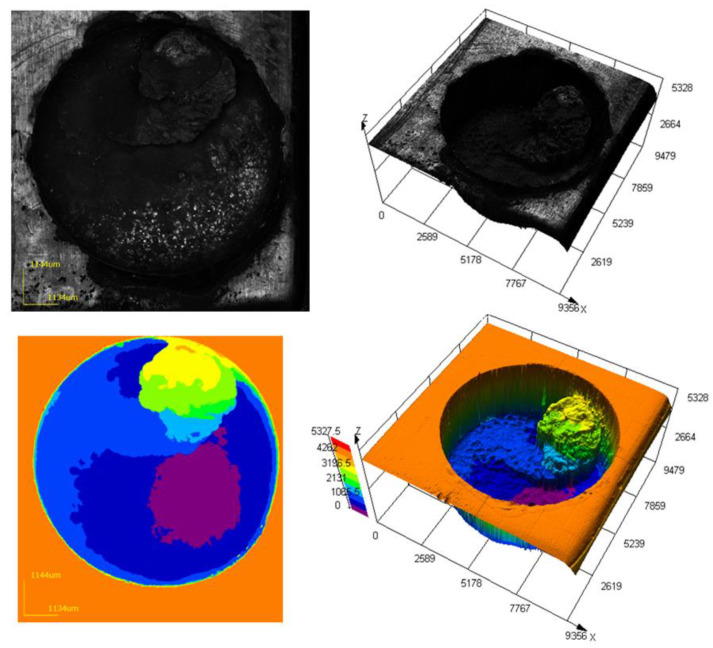
Three-dimensional image of the surface after processing to a depth of 15 mm.

## Data Availability

Not applicable.

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
