# Peer review of "Modeling of EDM Process Flushing Mechanism"

_materials, 2023, doi:10.3390/ma16114158_

Round 1
Reviewer 1 Report
The study was performed to determine the optimum flushing condition for electrical discharge machining of functional materials. Copper is used as a tool electrode to machine functional materials. The optimum flushing flows are analyzed theoretically and also validated by an experimental investigated. It was observed that while machining the functional materials to the depth of 10 mm or more, the turbulence fluid flow dominates when nozzle angles are 45Ëš and 75Ëš, consequently drastically affecting the quality of flushing and the performance of the EDM. The optimum flushing at deep hole EDM process minimizes the occurrence of debris deposition on tool electrode, thus facilitating stable machining of functional materials. This study is relatively meaningful. However, this paper could not be published in the present state, since one issue is needed to address.
1. The title of Figure 4 is wrong.
Author Response
Dear Reviewer,
I am grateful for the helpful and interesting comments by you.
We have revised the article.
We tried to make it better.

Reviewer 2 Report
Dear Editor and Authors,
In review, I received a manuscript draft entitled “Modeling of EDM process flushing mechanism” considered for publication in the MDPI journal “Materials”.
The manuscript explores the EDM process flushing mechanism. The authors performed simulations and verified them with experimental tests. The research question is focused on turbulence fluid flow developed when machining depths deeper than 10 mm, and the nozzle angle of 15° provides a suitable solution to this challenge.
Here are my suggestions and comments to improve the manuscript before publication.
L20: “these materials” => Please specify in the Introduction to which materials are you referring, and describe why is complicated (or reference). This is why the Introduction is intended for.
L22: please include the type of material (“carbon fibre reinforced functional composite materials”) in the title/abstract/keywords. It will help the reader to comprehend what the article is about.
L90: explain the abbreviation in full when are first time used (here ET).
L 93-94 is … considered as – (dash?) please revise. Sentence in L94 appears not finished – please rewrite (Upon receipt of this blank, the possibility of conducting a study of harmful gas impurities contained in the atmosphere.)
Materials and Methods section: please follow the standard notation when describing the devices used: full name (abbreviation, model, producer, country) or similar; update everywhere where applies. Figures 1-3 can be merged in one composite figure.
Results: Please merge Fig 6-14 into few logical composite figures and mark differences.
L204: the text is referring to functional materials – are we still talking about (as stated in introduction) carbon fibre reinforced functional composite materials? Please specify to avoid ambiguity. How the physical properties of this material were set? From references?
From experimental proof I would expect at least the optical microscopy of the surfaces at different depts to validate the removal of debris. Can you include this in the result, to support the main findings? Also, visualise and indicate the effect of secondary discharges, probably resulting in the machined surface discontinuities.
Conslusions, L235: some of the sentences appear half-finished, such as “Also, the possibility of secondary discharges originates.” Please revise the whole text carefully, and avoid duplications and orphaned sentences.
L236 statement of experimental confirmation of sludge sticking to the surface has to be supported somehow; I guess with some kind of microscopy. Currently, you show only a consequence of this on Fig15, but the experimental proof is something different.
L238 – here, the functional materials are referred to as titanium VT6 obtained by plasma cladding with a thermal cycle (for the first time). Therefore – what was the actual material investigated in this study – please specify (for simulation, and for experimental investigation). Be clear and concise.
Whole text: Please proofread the text for missing and multiple spaces, typos etc., including figure subtitles.
Overall, this could be publishable research. The authors have to specify undoubtedly which materials they were simulating and experimentally verifying, otherwise the manuscript cannot be published in this form.
Author Response

(The authors gave the same response as above.)

Reviewer 3 Report
This paper models the EDM process of flushing mechanism, which has potential application value in engineering. In order to meet the requirements of high-quality publication of the journal, it is recommended to consider the following suggestions.
1) There is no key quantitative data in the abstract.
2) Introduction Section needs to be rewritten. For example, small paragraphs, typography, and other issues.
3) Why select ANSYS CFX 20.1 10 software?
3) "Figure 4. Meshmodelforcalculations" is type wrong.
4) Why is there no subsection in the third Section? The logic is very poor.
5) The mehtod proposed in this paper needs to be compared with the previous literature, otherwise it cannot reflect innovation.
6) The mehtod proposed in this paper needs to be verfied by experiments.
7) The Discussion Section needs a separate section.
8) There is no quantitative data in the Conclusion Section.
9) "Funding: " in red color, why?
10) There are few references in the last three years.
Author Response

(The authors gave the same response as above.)

Round 2
Reviewer 2 Report
Dear Authors and Editor,
The revised manuscript adequately incorporates all the suggested revisions. The only suggestion that I would have is to carefully revise the added text in the Abstract - it appears added in haste, stating data that are hard to comprehend. After all, the Abstract is the first thing the reader will read.
All the best.
I have no further comments, just please revise few new sentences in the Abstract.
Author Response
The team of authors expresses deep respect for your work!
With your help, the article has become much better!

Reviewer 3 Report
The authors have addressed all my concerns.
Author Response

(The authors gave the same response as above.)
